# No Impact of NRAS Mutation on Features of Primary and Metastatic Melanoma or on Outcomes of Checkpoint Inhibitor Immunotherapy: An Italian Melanoma Intergroup (IMI) Study

**DOI:** 10.3390/cancers13030475

**Published:** 2021-01-26

**Authors:** Michele Guida, Nicola Bartolomeo, Pietro Quaglino, Gabriele Madonna, Jacopo Pigozzo, Anna M. Di Giacomo, Alessandro M. Minisini, Marco Tucci, Francesco Spagnolo, Marcella Occelli, Laura Ridolfi, Paola Queirolo, Ivana De Risi, Davide Quaresmini, Elisabetta Gambale, Vanna Chiaron Sileni, Paolo A. Ascierto, Lucia Stigliano, Sabino Strippoli

**Affiliations:** 1Rare Tumors and Melanoma Unit, IRCCS Istituto Tumori “Giovanni Paolo II”, 70124 Bari, Italy; i.derisi@oncologico.bari.it (I.D.R.); davide.quaresmini@hotmail.it (D.Q.); strippoli.sabino@libero.it (S.S.); 2Department of Biomedical Sciences and Human Oncology, University of Bari, 70124 Bari, Italy; nicola.bartolomeo@uniba.it; 3Department of Medical Sciences, Dermatologic Clinic, University of Turin, 10126 Turin, Italy; pietro.quaglino@unito.it (P.Q.); lucia.stigliano@edu.unito.it (L.S.); 4Department of Melanoma, Cancer Immunotherapy and Development Therapeutics, Istituto Nazionale Tumori IRCCS Fondazione “G. Pascale”, 80131 Napoli, Italy; gabriele.madonna@yahoo.it (G.M.); p.ascierto@istitutotumori.na.it (P.A.A.); 5Melanoma Oncology Unit, Veneto Institute of Oncology IOV-IRCCS, 31033 Padova, Italy; jacopo.pigozzo@iov.veneto.it (J.P.); vanna.chiarion@iov.veneto.it (V.C.S.); 6Center for Immuno-Oncology, Medical Oncology and Immunotherapy, Department of Oncology, University Hospital of Siena, 53100 Siena, Italy; a.digiacomo@ao-siena.toscana.it (A.M.D.G.); gambaleelisabetta@gmail.com (E.G.); 7Department of Oncology, ASUFC University Hospital, 33100 Udine, Italy; alessandro.minisini@asuiud.sanita.fvg.it; 8Medical Oncology Unit, IRCCS Istituto Tumori “Giovanni Paolo II”, University of Bari Aldo Moro, 70124 Bari, Italy; marco.tucci@uniba.it; 9Skin Cancer Unit, IRCCS Ospedale Policlinico San Martino, 16132 Genova, Italy; francesco.spagnolo85@gmail.com; 10Azienda Ospedaliera Santa Croce e Carle di Cuneo SC Oncologia, 12100 Cuneo, Italy; marcellaoccelli@gmail.com; 11Department of Oncology, IRCCS Istituto Romagnolo per lo Studio dei Tumori (IRST) “Dino Amadori”, 47014 Meldola, Italy; laura.ridolfi@irst.emr.it; 12Division of Melanoma Sarcoma and Rare Tumors, IEO European Institute of Oncology IRCCS, 20141 Milan, Italy; paola.queirolo@ieo.it

**Keywords:** melanoma, NRAS mutation, immunotherapy, checkpoint inhibitors

## Abstract

**Simple Summary:**

Neuroblastoma RAS Viral Oncogen Homolog (NRAS) mutant melanoma is usually considered more aggressive and more responsive to checkpoint inhibitor immunotherapy (CII) than NRAS wildtype. We retrospectively recruited 331 metastatic melanoma patients treated with CII as first line: 162 NRAS-mutant/BRAF wild-type and 169 wt/wt. No substantial differences were observed among the two cohorts regarding the melanoma onset and disease-free interval. Also, overall response to CII, progression-free survival and overall survival were similar in the two groups. Therefore, our data do not show increased aggressiveness and higher responsiveness to CII in NRAS-mutant melanoma. The controversy in the published data could be due to different patient characteristics and treatment heterogeneity. We believe our data adds evidence to clear up these controversial issues.

**Abstract:**

Aims: It is debated whether the NRAS-mutant melanoma is more aggressive than NRAS wildtype. It is equally controversial whether NRAS-mutant metastatic melanoma (MM) is more responsive to checkpoint inhibitor immunotherapy (CII). 331 patients treated with CII as first-line were retrospectively recruited: 162 NRAS-mutant/BRAF wild-type (mut/wt) and 169 wt/wt. We compared the two cohorts regarding the characteristics of primary and metastatic disease, disease-free interval (DFI) and outcome to CII. No substantial differences were observed between the two groups at melanoma onset, except for a more frequent ulceration in the wt/wt group (*p* = 0.03). Also, the DFI was very similar in the two cohorts. In advanced disease, we only found lung and brain progression more frequent in the wt/wt group. Regarding the outcomes to CII, no significant differences were reported in overall response rate (ORR), disease control rate (DCR), progression free survival (PFS) or overall survival (OS) (42% versus 37%, 60% versus 59%, 12 (95% CI, 7–18) versus 9 months (95% CI, 6–16) and 32 (95% CI, 23–49) versus 27 months (95% CI, 16–35), respectively). Irrespectively of mutational status, a longer OS was significantly associated with normal LDH, <3 metastatic sites, lower white blood cell and platelet count, lower neutrophil-to-lymphocyte (N/L) ratio. Our data do not show increased aggressiveness and higher responsiveness to CII in NRAS-mutant MM.

## 1. Introduction

The medical treatment of metastatic melanoma (MM) has recently been significantly improved by the identification of specific genetic alterations, such as BRAF, NRAS and cKIT mutations. Approximately 50% of melanomas harbour BRAF mutations, whereas NRAS and KIT mutations are found in approximately 20%, and 2–3% of cases, respectively [1]. Metastatic BRAF-mutant disease can be targeted with specific BRAF inhibitors in combination with MEK inhibitors to achieve response rates of 70–80% with a median progression-free survival (PFS) of 11–15 months and a median overall survival of over 2 years [2].

In the last few years, immunotherapy has also played a primary role in the treatment of MM due to the availability of new drugs, such as monoclonal antibodies directed toward the checkpoint molecules CTLA-4 (cytotoxic T lymphocyte antigen-4) and PD-1 (programmed death antigen 1) or its ligand. This approach is able to reverse the immunosuppressive status and restart a potent antitumoural immune response. The anti-CTLA-4 antibody ipilimumab is able to induce a response rate of approximately 15% with approximately 20% of patients being long-term responders [3]. More recently, the anti-PD1 drugs nivolumab and pembrolizumab have been proven a higher response rate of approximately 40% in treatment-naïve patients, with the majority of responses being durable [4].

It has been reported that melanoma with activating NRAS mutations has a more aggressive course of disease compared to NRAS wild-type melanoma [5,6,7]. Nevertheless, these findings were not confirmed by other authors [8]. In relation to treatment, no specific targeted therapies are available for NRAS mutations. Moreover, the use of MEK inhibitors as experimental drugs was found to not significantly improve PFS or OS compared to standard chemotherapy [9]. Therefore, the standard treatment of NRAS-mutant patients is currently the same as for BRAF wild-type melanoma; anti-PD-1 based immunotherapy is the first-line therapy, and the second-line therapy includes anti-CTLA-4, cytotoxic chemotherapy, or drugs in experimental trials.

Currently, no validated biomarkers are able to predict clinical responses to checkpoint inhibitors. Additionally, PD-L1 expression, likely associated with a higher response rate, is weakly correlated with better survival [10]. In this regard, it is unclear whether specific driver mutations influence immunotherapy outcomes. It is equally controversial whether the NRAS mutation is associated with a higher responsiveness to immunotherapy [11,12,13].

To verify these data, we explored the effect of NRAS mutations on melanoma characteristics and on checkpoint inhibitor immunotherapy (CII) outcomes in a large population. To this purpose, we retrospectively compared two cohorts of patients homogeneously treated with CII as the first-line therapy. The first cohort included NRAS-mutant/BRAF wild type patients (mut/wt); the second cohort included BRAF- and NRAS-wild type patients (wt/wt).”

## 2. Results

### 2.1. Demographics

A total of 331 patients were recruited from 11 referral centers in Italy, including 162 mut/wt patients and 169 wt/wt patients. Patients were treated in a time period of 92 months (from January 2011 to August 2019).

We found very similar patient characteristics between the two groups at study accrual and a good balance of prognostic factors. The most common NRAS mutations included Q61K (60 patients, 37%) and Q61R (56 patients, 35%) (Table 1).

### 2.2. Characteristics of Primary Melanoma, Disease-Free Interval and Metastatic Disease

At melanoma onset, no significant differences were found (Table 2). The median duration of DFI was very similar in the two cohorts, with 15.4 months (range 4–36) for mut/wt and 15 months (range 3–37) for wt/wt. 

Regarding metastatic disease, we found no differences in age, LDH levels, number of metastatic sites, or ECOG PS. In contrast, a significant difference was found in the site of metastases, with more frequent lung and brain metastases in the wt/wt group (*p* < 0.01 and *p* = 0.01, respectively) and soft tissue and lymph node metastases in the mut/wt group (*p* = 0.07 and *p* = 0.09, respectively). Additionally, progression to the brain was higher in the wt/wt group (*p* < 0.01). Peripheral blood cellular evaluation was assessed in 151 patients in the mut/wt cohort and in 153 patients in the wt/wt cohort. None of these parameters showed significant differences between the two cohorts (Table 2 and Table 3).

### 2.3. Treatments Used for Metastatic Disease

In the wt/wt cohort, 35 patients were treated with ipilimumab, 132 with an anti-PD-1 antibody (nivolumab, pembrolizumab) and 2 patients with the combination. In the mut/wt cohort, 45 patients received ipilimumab, 114 patients received anti-PD-1 and 3 received the combination. All treatments were administered until disease progression or unacceptable toxicity.

A total of 56 patients (35%) in the NRAS mutant group and 61 patients (36%) in the wt/wt group received a second-line systemic treatment (anti-PD-1 after ipilimumab in 24 patients, ipilimumab after anti-PD-1 in 9 patients, chemotherapy in 10 patients, others in 13 patients; anti-PD-1 after ipilimumab in 23 patients, ipilimumab after anti-PD-1 in 16 patients, chemotherapy in 13 patients, others in 9 patients, respectively). Moreover, 17 patients (10%) in the NRAS mutated group and 10 (6%) in the wt/wt group received a third-line systemic therapy.

### 2.4. Response to Treatment

We assessed the response to therapy in all patients included in the study. We observed no benefit in terms of ORR in the mut/wt group compared to the wt/wt group (42% versus 37%; *p* = ns). DCR (CR plus PR, plus SD lasting more than 6 months) was similar between the two groups (60% and 59%, respectively; *p* = ns). These findings were observed for both ipilimumab and anti-PD-1 treatments (Table 4).

Ipilimumab plus nivolumab was used globally in only 5 patients, namely, 3 mut/wt patients and 2 wt/wt patients; two CR and 1 PR were reported in the mut/wt group; one CR and 1 SD lasted 6 months in the wt/wt group.

Finally, response was analyzed according to NRAS mutation subtypes and no differences were found (Table 5).

In the univariate analysis, a better DCR was associated with the use of anti-PD-1 as the first-line therapy (*p* < *0*.001), normal LDH levels (*p* = 0.03), number of metastatic sites <3 (*p* < 0.01), lower N/L ratio (*p* = 0.05), and a lower platelet count (*p* < 0.01). In the multivariate analysis, statistical significance was maintained by the use of anti-PD-1 as the first-line therapy (*p* < 0.001), <3 metastatic sites (*p* < 0.01), and a lower platelet count (*p* = 0.03). Moreover, a better response was associated with head and neck site versus the trunk, head and neck site versus lower limbs, and upper limb site versus lower limbs. Finally, we found statistical significance regarding the presence of NRAS mutations (*p* = 0.03) (Table 6).

The discrepancy of the effects of NRAS mutation status on DCR between the univariate and multivariate analyses can be explained by the following: in the multivariate analysis, the sample size was reduced to 267 patients due to missing values in some parameters, and the presence of effect modifiers influencing the relationship between the presence/absence of the mutation and DCR [14].

### 2.5. Progression-Free Survival and Overall Survival

The median duration of PFS was 11 months (95% CI, 7–14), with 12 months (95% CI, 7–18) for the mut/wt group and 9 months (95% CI, 6–16) for the wt/wt group (*p* = 0.51). In the cohorts of patients treated with ipilimumab monotherapy, PFS in the mut/wt group was 4 months (95% CI, 3–6) compared with 3 months (95% CI, 3–6) in the wt/wt group (adjusted *p* = ns). For patients treated with anti-PD-1, PFS was 15 (95% CI, 11–29) months and 16 (95% CI, 8–24) months in the mut/wt and wt/wt groups, respectively (adjusted *p* = ns) (Figure 1 and Figure 2).

The 3 patients in the mut/wt group who were treated with ipilimumab plus nivolumab did not progress after 30, 34 and 39 months. In contrast, in the wt/wt group, the patient obtaining SD died after 8 months, and the other responder patient was progression free after 41 months.

In the univariate analysis, a longer PFS was associated with the use of anti-PD-1 as the first-line therapy (*p* < 0.0001), normal LDH levels (*p* < 0.01), and <3 metastatic sites (*p* < 0.01). A positive trend near statistical significance was also found for a lower N/L ratio (*p* = 0.06).

In the multivariate analysis, a better PFS was associated with the presence of NRAS mutations (*p* = 0.05), anti-PD-1 as first line therapy (*p* < 0.0001), number of metastatic sites <3 (*p* < 0.01), and a higher platelet count (*p* = 0.04). Moreover, PFS was also significantly correlated with the site of primary melanoma (*p* = 0.03). A trend of a better PFS was found for sex, with females having a better PFS than males (*p* = 0.05). Finally, NRAS mutations were found at the limits of statistical significance (*p* = 0.05) (Table 7).

Regarding overall survival (OS), we recorded a median OS of 28 months (95% CI, 23–35) with 32 months (95% CI, 23–49) in the mut/wt group and 27 months (95% CI, 16–35) in the wt/wt group (*p* = ns). In patients treated with ipilimumab monotherapy as the first-line therapy, the OS was 26 (95% CI, 14–48) and 15 (95% CI, 11–35) months (*p* = ns), whereas in patients treated with anti-PD-1, the OS was 32 months (95% CI, 22-NA) and 27 months (95% CI, 18–42) in the mut/wt and wt/wt groups, respectively (*p* = ns) (Figure 2).

In the univariate analysis, a longer OS was associated with normal LDH levels (*p* < 0.001), <3 metastatic sites (*p* < 0.01), lower white blood cell count (*p* = 0.03), lower platelet count (*p* = 0.01), lower N/L ratio (*p* < 0.01), and site of the onset melanoma (*p* = 0.03) (Table 8).

In the multivariate analysis, LDH level maintained the strongest statistical significance (*p* < 0.01). The following were also significant: White blood cell count (*p* = 0.04), platelet count (*p* < 0.01) and site of primary melanoma (*p* = 0.01). Female sex was found near statistical significance (*p* = 0.05). NRAS mutation resulted not significant (*p* = 0.07).

Finally, regarding performance status, despite the limitations due to the strong imbalance of the ECOG PS 0–1 versus 2–3 in both groups, in the multivariate analysis we found statistically shorter PFS and OS for patients with PS 2–3 compared to those with PS 0–1.

All 3 patients in the NRAS-mutant group who were treated with ipilimumab plus nivolumab were alive, with a median OS of 38+ months for the two patients who achieved a CR and 30+ months for the patient who achieved PR. In the wt/wt group, the OS was 8 months for the SD patient and 44+ months for the CR patient.

## 3. Discussion

It is controversial whether NRAS mutations in melanoma lead to a distinct clinicopathological phenotype and increased aggressiveness compared to NRAS wild-type melanomas. It is equally controversial whether NRAS mutations are associated with a higher responsiveness to immunotherapy.

In relation to primary melanoma, NRAS mutations have been associated with thicker lesions, higher mitotic rates, nodular primary subtype, and nodal relapse compared to BRAF-mutated and wild-type melanoma [5,15]. Other authors, however, did not confirm these findings [8]. Even Carlino et al. [16], in 193 patients, (92 BRAF-mutant, 39 NRAS-mutant and 62 wild-type patients) reported a longer disease-free interval in NRAS-mutant patients (49 months) than in wild type (27,9 months) and BRAF-mutant (35 months) patients [16].

Additionally, in metastatic disease, NRAS-mutant melanoma has been reported to be more aggressive than NRAS wild-type melanoma, with poorer survival and a higher percentage of patients developing brain metastases [5,6,7]. Nevertheless, other authors did not find a difference in melanoma-specific survival dependent on mutations [8].

In our population, we reported no substantial differences between the two cohorts regarding the characteristics of primary melanoma, and a very similar DFI (15.4 months for the mut/wt group and 15 months for the wt/wt group).

Additionally, for metastatic disease, we found slight differences between the two groups. Statistical significance was achieved for lung and brain location and major brain progression were more frequent in the wt/wt group.

As previously mentioned, some studies reported a better response of metastatic NRAS-mutant melanoma to immunotherapy with high-dose interleukin-2 [11] or checkpoint inhibitors [12]. Moreover, the analysis of the phase III NEMO study comparing the MEK inhibitor binimetinib to standard chemotherapy in NRAS-mutant patients noted that patients previously treated with immunotherapeutic agents had a longer PFS and response compared to the overall population [9]. Some authors argued that this advantage could be attributed to late-onset benefits from previous immunotherapy [17].

These benefits have been explained by the presence of a higher mutational burden [18] and higher expression of PDL-1 in NRAS-mutant melanomas [12]. Nevertheless, this latter hypothesis is based on a fairly weak evidence derived from an exploratory analysis carried out on only 39 samples [12]. In contrast, preclinical data from 51 melanoma cell lines reported no correlation between mutational status and PDL-1 expression [19].

In relation to clinical outcomes, Johnson and colleagues retrospectively analyzed 229 MM patients (60 NRAS-mutant, 53 BRAF-mutant and 116 wt/wt patients) that were treated with first-line immunotherapies, including interleukin-2, ipilimumab and anti-PD-1/PD-L1. These authors reported a global response rate of 28% in NRAS mutant MM versus 16% in MM of other genotypes (*p* = 0.04), with a clinical benefit of 50% versus 31% (*p* < 0.01) and a PFS of 4.1, versus. 2.9 months (*p* = 0.09), respectively. The benefit was particularly marked in the NRAS-mutant cohort treated with anti-PD-1/PD-L1 (73% versus 35%) [12]. Likewise, a trend towards a better OS was observed in NRAS mutant (19.5 months versus 15.2 months) even if a worse prognosis was observed in NRAS mutant that not responded to immunotherapy.

In another retrospective study of 101 patients treated with anti-CTLA-4 immunotherapy, no association between BRAF/NRAS mutational status and OS was found [20].

More recently, Kirchberger et al [13]. reported data from 236 NRAS-mutant and 128 NRAS wild-type patients, 48 of whom harbored BRAF mutations. Moreover, approximately half of the patients in both groups were treated with systemic treatment prior to checkpoint blockade, including chemotherapy and kinase inhibitors (MEK inhibitors in 19% of NRAS-mutant patients and BRAF/MEK-inhibitors in 18% of BRAF-mutant patients). These authors reported similar response rates to checkpoint inhibitors in the two cohorts, with PFS of three months versus four months in the NRAS wild type and NRAS mutant groups, respectively. Moreover, the median OS was significantly lower in NRAS mutant group compared to the NRAS wild-type group (21 versus 33 months, *p* = 0.034). In this regard, this advantage was influenced by the use of an active treatment with anti-BRAF/anti-MEK in BRAF-mutant patients included in this group. The OS in the NRAS-mutant group was 12 months for ipilimumab, 18 months for anti-PD-1, and 32 months for ipilimumab plus anti-PD-1. Finally, therapy with oral anti-MEK before or after checkpoint inhibitors in NRAS-mutant patients resulted in a survival benefit [13].

In our population, we also found no substantial differences between the two groups in terms of response rate and DCR (42% and 60% in mut/wt versus 37% and 59% in wt/wt). Moreover, no difference was found between the main NRAS subtypes. Interestingly, we found a major benefit for the mut/wt group receiving ipilimumab compared to wt/wt group (36% versus 17%; *p* = 0.07). In the univariate analysis, a better DCR was associated with the use of anti-PD-1 as the first-line therapy (*p* < 0.001), normal LDH levels (*p* = 0.03), <3 metastatic sites (*p* < 0.01), lower platelet count (*p* < 0.01), and lower N/L ratio (*p* = 0.05). In the multivariate analysis, statistical significance was associated with anti-PD-1 used as the first-line therapy (*p* < 0.001), <3 metastatic sites (*p* < 0.01), site of melanoma onset (head and neck were better than trunk and lower limbs, and upper limbs were better than lower limbs), lower platelet count (*p* = 0.03), and presence of NRAS mutation (*p* = 0.05). As previously mentioned, the discrepancy in the effect of NRAS mutation on DCR among univariate and multivariate analysis is due to both the reduction of the sample size in the multivariate analysis and the interaction between the presence/absence of mutation and other parameters considered in the model [14]. 

Regarding the performance (PS) status, despite the limitations due to the strong imbalance of the ECOG PS 0–1 versus 2–3 in both groups, we found a statistically lower response in patients with PS 2–3 compared to those with PS 0–1.

Additionally, PFS was similar in the two groups (11 months in the global population, with 12 months in the mut/wt group and 9 months in the wt/wt group) with four months and three months for ipilimumab and 15 months. and 16 months for anti-PD-1, respectively. Anti-PD-1 used as the first-line therapy, normal LDH levels, <3 metastatic sites, PS, and site of primary melanoma (head and neck was better than unknown site, trunk or lower limbs) were positively correlated with PFS. In the multivariable analysis, a better PFS was associated with the presence of NRAS mutations and a higher platelet count. Interestingly, a trend for a better PFS was also found for females versus males.

Regarding the anti-PD-1 group, we reported a PFS longer than in large, controlled studies which is of approximately 5–7 months [4,21]. In our opinion, this longer PFS could be explained in two ways: (1) The presence of more favorable prognostic factors in our patient population with respect to controlled studies (elevated LDH approximately 30% versus over 35% and M1c stage of approximately 45% versus approximately 60%) [4,21]. (2) A delayed time in response assessment that is less generally rigorous in clinical practice than in controlled studies.

Regarding survival, we reported a global median OS of 28 months, with 32 months for the mut/wt group and 27 months for the wt/wt group (*p* = ns). Interestingly, ipilimumab monotherapy was associated with a better OS in mut/wt patients than in wt/wt patients, whereas in anti-PD-1-treated patients, the OS was 32, and 27 months, respectively (*p* = ns). These data are quite similar to those reported with the anti-PD-1 pembrolizumab and nivolumab in large, controlled studies [4,21].

In the univariate analysis, we found a significantly longer OS associated with normal LDH, PS, number of metastatic sites <3, lower white blood cell and platelet counts, N/L ratio, and site of primary onset with head and neck better than trunk and lower limbs, and upper limbs better than lower limbs. In the multivariate analysis, normal LDH levels maintained the strongest statistical significance (*p* < 0.01) together with PS (*p* < 0.004), site of primary onset (*p* = 0.01), lower white blood cell counts (*p* = 0.04) and platelet counts (*p* < 0.01). Female sex was close to statistical significance (*p* = 0.05). 

For PFS and OS, we also found no differences regarding NRAS mutations between genotype subgroups.

Basal neutrophils and the N/L ratio were recently reported to be prognostic in MM patients receiving ipilimumab [22] or anti-PD-1 [23,24,25]. Also, thrombocytosis and a high platelet-to-lymphocyte ratio present in 20–60% of cancer patients, have been reported as independent prognostic factors in numerous solid tumours [26,27,28], but no data are available regarding their possible predictive/prognostic role in patients treated with checkpoint inhibitors.

We found a better DCR associated with a lower N/L ratio (*p* = 0.05) and lower platelet count (*p* < 0.01), with platelets also maintaining statistical significance in the multivariate analysis (*p* = 0.03). Also, a longer OS was found to be associated with a lower white blood cell count (*p* = 0.03), lower platelet count (*p* = 0.01), and lower N/L ratio (*p* < 0.01), with white blood cell count and platelet maintaining their strong significance in the multivariate analysis (*p* = 0.04, *p* < 0.01, *p* = 0.04, respectively). Due to the ease in acquiring this information, these parameters could be utilized as potential biomarkers for stratification in clinical trials and for use in clinical practice during checkpoint inhibitor immunotherapy.

In summary, our data do not support increased aggressiveness of NRAS mutant melanoma. Additionally, CII used as first-line therapy was equally effective in NRAS-mutant melanoma compared with NRAS wild-type melanoma. The controversy in the published data could be due to heterogeneity both in patient characteristics and in treatments.

The strength of our study are the homogeneity of the two cohorts of patients studied, the use of checkpoint inhibitors as the first-line therapy in all patients, the good balance of the two groups regarding prognostic parameters, the accrual in a limited period of time, and the exclusion of the BRAF-mutant population. 

However, some weaknesses remain and should be highlighted, including the retrospective nature of the study, the heterogeneity of the ways and timing of evaluation of the response to immunotherapy, and the presence of missing data for some patients in the analyses.

## 4. Materials and Methods 

### 4.1. Patients and Study Design

Patients were treated in a time period of 92 months (from January 2011 to August 2019).

First, we verified the differences between the two groups regarding the characteristics of primary tumour presentation, disease-free interval (DFI), and metastatic disease characteristics. Then, we evaluated the response rate, progression-free survival (PFS) and overall survival (OS) to CII in the two cohorts.

We excluded patients with BRAF mutations to avoid the confounding impact of anti-BRAF/anti-MEK-targeted therapy, which is the highly efficacious standard first-line therapy for patients harboring BRAF mutations. To make the two groups even more homogeneous, mucosal and ocular melanoma were also excluded.

Patients were identified from the databases of their respective Centres. The main inclusion criteria were diagnosis of metastatic cutaneous melanoma or melanoma of unknown origin, first-line treatment with checkpoint inhibitors and presence of evaluable disease. Patients had to be treated for at least 1 month with checkpoint inhibitor immunotherapy.

The main recorded patient characteristics included sex, age at diagnosis, origin and characteristics of primary cancer, previous systemic adjuvant therapy, BRAF/NRAS genotype, age at metastasis, Eastern Cooperative Oncology Group (ECOG) performance status at the beginning of therapy, M stage and sites of metastases, presence of brain metastases, lactate dehydrogenase (LDH) level at metastatic disease, and subsequent therapies after first-line treatment. 

Objective tumour responses were assessed by investigators using instrumental analysis, such as computed tomography or magnetic resonance imaging.

Each center, after the approval of its ethical committee, recorded the clinical data of their patients in an electronic local database. Then, all databases were collected in the central database at Istituto Tumori “Giovanni Paolo II”, Bari, Italy.

### 4.2. Genetic Analysis

All patients underwent genetic analysis to verify the presence of BRAF or NRAS mutations. Mutation testing was performed from archival paraffin-embedded formalin-fixed samples from metastatic disease (30% loco-regional lymph nodes metastases, 70% non-lymph nodes metastases). Based on the method used in the Centers that participated in the study, molecular profiling was performed by Real-Time PCR or NGS (75% and 25%, respectively). Patients without identified mutations in NRAS or BRAF were classified as wild type and patients with BRAF mutations were excluded from the study.

### 4.3. Treatment and Clinical Outcomes

All 331 patients were treated with checkpoint inhibitor immunotherapy as the first-line treatment, including the anti-PD-1 antibodies nivolumab and pembrolizumab or the anti-CTLA-4 antibody ipilimumab. All drugs were administered intravenously according to the standard doses and schedules.

Clinical responses were assessed and reported as complete response (CR), partial response (PR), stable disease (SD), and progressive disease (PD) based on the Response Evaluation Criteria in Solid Tumor version 1.1 [29]. We also evaluated the overall response rate (ORR) as CR plus PR, and the disease control rate (DCR) was defined as CR plus PR plus SD lasting 6 months or more. Patients with SD less than 6 months were included in the PD group. Responses to subsequent therapies were also assessed. DFI, PFS and OS were also evaluated in the two groups of patients.

### 4.4. Statistical Analysis

The DFI was defined as the time to recurrence from first diagnosis. PFS was defined as the time from first immune therapy treatment to first progression or death. OS was calculated by the date of first immune therapy treatment to date of death for any reason. Patients who were alive at the last date of follow-up were censored for OS. Patients alive and progression-free were censored for PFS.

The Shapiro-Wilk test was used to determine whether the continuous variables (age at diagnosis, age at metastatic disease, peripheral white blood cell count, platelet and lymphocyte count, neutrophil/lymphocyte ratio at metastatic disease) showed a normal distribution. Then, these variables were expressed as the median and interquartile range (IQR). A nonparametric Mann-Whitney U test was used to compare these variables between the NRAS mutated/nonmutated groups. The association between qualitative variables and NRAS mutated/non-mutated groups was assessed by the chi-squared test or exact Fisher test as necessary. For parameters with more than two levels, the *p* value of the pairwise comparisons was adjusted by the Bonferroni test. Through a univariate logistic model, we tested the effect of each variable on the probability of a positive DCR. The log-rank test was used to compare between the Kaplan-Meier survival curves in NRAS mutated/non-mutated groups for both PFS and OS. The effect of each parameter on PFS and OS was assessed using the Cox regression model, and the proportional hazard assumptions for the Cox model were evaluated. All variables were included in the multivariable logistic regression model [30] to evaluate the DCR. Similarly, in the multivariable Cox regression model, all variables were included to evaluate PFS and OS outcomes. Stepwise selection, using the Akaike Information Criterion (AIC), was used to estimate the final models. The results of the logistic model and the Cox model are expressed respectively by the odds ratios (OR), hazard ratios (HR), 95% Wald confidence intervals, and the *p* values of the Wald chi-square tests.

All tests of statistical significance were two-tailed, and *p*-values less than 0.05 were considered statistically significant. Statistical analysis was performed in R statistical software (version 3.5.3, R Core Team, Vienna, Austria) using “RcmdrPlugin.EZR” package [31].

## 5. Conclusions

Our data do not support increased aggressiveness and higher responsiveness to CII of NRAS-mutant melanoma with respect to NRAS/BRAF wild type. The controversy in the published data could be due to different patient characteristics and treatment heterogeneity used. We believe our paper adds data to clear up these controversial issues.

## Figures and Tables

**Figure 1 cancers-13-00475-f001:**
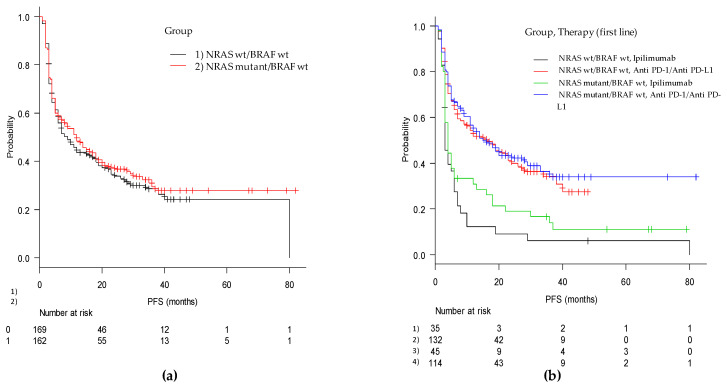
Kaplan-Meier curves of progression free survival from first-line immune-based therapy in (**a**) the mut/wt and wt/wt cohorts and in (**b**) the two groups according to checkpoint inhibitor utilized (anti-PD-1 and anti-CTLA4).

**Figure 2 cancers-13-00475-f002:**
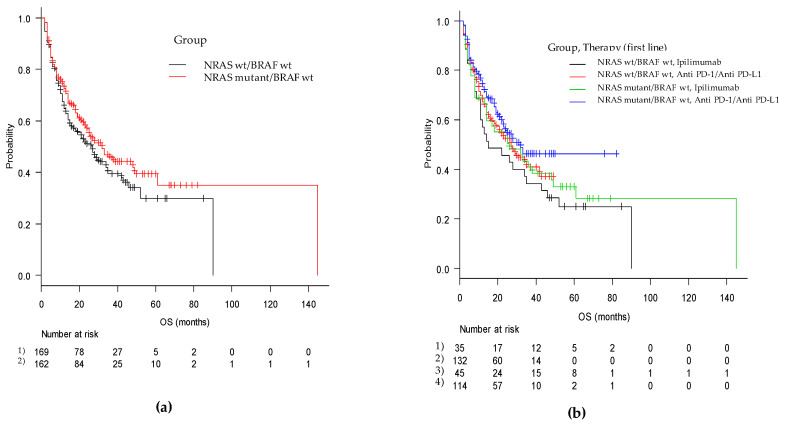
Kaplan-Meier curves of overall survival from first-line immune-based therapy in (**a**) the mut/wt and wt/wt cohorts and in (**b**) the two groups according to checkpoint inhibitor utilized (anti-PD-1 and anti-CTLA4).

**Table 1 cancers-13-00475-t001:** Summary of clinical characteristics and treatment selection of the study cohort (*n* = 331 patients).

Clinical Features	NRAS Mutant/BRAF wt *n* (%)	NRAS wt/BRAF wt *n* (%)	*p* Value *
Total	162	169	
Origin of melanoma			
Cutaneous Unknown	137 (85) 25 (15)	147 (87) 22 (13)	0.53
Gender			
Female Male	58 (36) 104 (64)	69 (41) 100 (59)	0.35
Adjuvant therapy			
Yes No	22 (14) 140 (86)	13 (8) 156 (92)	0.08
Therapy (first line)			
Anti PD-1 Ipilimumab Anti PD-1 + Ipilimumab	114 (70) 45 (28) 3 (2)	132 (78) 35 (21) 2 (1)	0.27
Therapy (second line)			
Yes No	56 (35) 106 (65)	61 (36) 108 (64)	0.77
Therapy (third line)			
Yes No	17 (10) 145 (90)	10 (6) 159 (94)	0.13

* Chi-square test.

**Table 2 cancers-13-00475-t002:** Main disease characteristics of the two cohorts at melanoma onset.

Disease Features	NRAS Mutant/BRAF wt	NRAS wt/BRAF wt	*p* Value
Age at diagnosis, years			0.86 *
Median (IQR)	63.4 (53.3–73.8)	65 (54.0–73.0)	
Site (%)			< 0.001 **
Head & neck	11 (7)	36 (21)	
Trunk	65 (40)	50 (30)	
Upper limbs	20 (12)	16 (9)	
Lower limbs	41 (25)	37 (22)	
Other	0	8 (5)	
Unknown	25 (15)	22 (13)	
Thickness (%)			0.76 **
pT1	10 (7)	10 (6)	
pT2	25 (17)	21 (13)	
pT3	35 (23)	38 (24)	
pT4	56 (37)	67 (42)	
Unknown	24 (16)	22 (14)	
Ulceration (%)			0.03 **
yes	71(44)	91 (54)	
No	51 (31)	32 (19)	
Unknown	40 (25)	46 (27)	
Lymph node status ^1^ (%)			0.48 **
Positive	70 (55)	68 (51)	
Negative	57 (45)	66 (49)	
DFI (months)			0.97 *
Median (IQR)	15.4 (4–36)	15 (3–37)	

(*) Mann-Whitney U test; (**) Chi- square test. (IQR): Inter Quartile Range, ^1^ Data available on 261patients.

**Table 3 cancers-13-00475-t003:** Patient and disease characteristics of the two cohorts at metastatic disease.

Clinical Features	NRAS Mutant/BRAF wt	NRAS wt/BRAF wt	*p* Value
Age at metastatic disease, years			
Median (IQR)	68 (54–76)	68.7 (56–76)	0.94 *
Serum LDH, *n* (%)			
Normal	92 (57)	90 (53)	0.67 **
Elevated	47 (29)	51 (30)	
Unspecified	22 (14)	28 (17)	
Blood cellular count (IQR)			
WBC (10^3^/μL)	6.7 (5.67–8.20)	6.8 (5.55–7.88)	0.77 *
Lymphocytes (10^3^/μL)	1.66 (1.31–2.06)	1.65 (1.31–2.1)	0.99 *
N/L	2.51 (1.89–3.60)	2.51 (1.67–3.6)	0.89 *
Platelets (10^3^/μL)	229 (194.7–297.2)	236.5 (181–288.2)	0.82 *
Site of disease, *n* (%) ^			
Skin/Soft tissue	90 (56)	77 (46)	0.07 **
Lymph node	112 (69)	102 (60)	0.09 **
Lung	75 (46)	103 (61)	<0.01 **
Liver	34 (21)	29 (17)	0.37 **
Brain	12 (7)	28 (17)	0.01 **
Bone	20 (12)	16 (9)	0.40 **
Other	33 (20)	16 (9)	<0.01 **
N. of metastatic sites, *n* (%)			
<3	94 (58)	108 (64)	0.27 **
≥3	68 (42)	61 (36)	
Stage at metastatic disease, *n* (%)			
III	1 (1)	2 (1)	<0.01 **
IVA	46 (28)	38 (22)	
IVB	41(25)	56 (33)	
IVC	64 (39)	45 (27)	
IVD	10 (6)	28 (17)	
Brain progression, *n* (%)			
Yes	17 (19)	35 (37)	<0.01 **
No	72 (81)	59(63)	
ECOG ^1^ PS, *n* (%)			
0–1	123 (76.4)	129 (76.3)	0.91 **
2	36 (22.4)	37 (21.9)	
3	2 (1.2)	3 (1.8)	

(IQR): Inter Quartile Range; (*) Mann-Whitney U test; (**) Chi-square test. ^ The % refers to the number of patients with that specific metastatic site of the whole group considered (several patients had more than one site of metastasis) (^1^) ECOG: Eastern Cooperative Oncology Group Performance Status; LDH: serum lactate dehydrogenase.

**Table 4 cancers-13-00475-t004:** Overall response and disease control rate in the 2 cohorts of patients.

Therapy	Response	NRAS Mutant/BRAF wt *n* (%)	NRAS wt/BRAF wt *n* (%)	*p* Value *
All patients	ORR	68 (42)	63 (37)	0.38
	DCR	97 (60)	100 (59)	0.90
Anti-PD-1	ORR	49 (43)	56 (42)	0.93
	DCR	78 (68)	88 (67)	0.77
Ipilimumab	ORR	16 (36)	6 (17)	0.07
	DCR	16 (36)	11 (31)	0.70

(*) Chi-square test, ORR: overall response rate; DCR: disease control rate including complete response, partial response and stable disease lasting more than 6 months.

**Table 5 cancers-13-00475-t005:** Distribution of NRAS mutation genotypes and clinical outcome (*n* = 162 patients).

NRAS Mutation	Frequency *n* (%)	ORR *n* (%)	*p* Value *	PFS, Median, Months (IQR)	*p* Value **	OS, Median, Months (IQR)	*p* Value **
Q61K	60 (37)	24 (40)	0.41	7.5 (4–20)	0.35	26 (14–48)	0.39
Q61R	56 (35)	25 (45)		12 (5–20)		28 (18-NA)	
Q61L	17 (10)	8 (47)		18 (3-NA)		NA (5-NA)	
Q61H	9 (6)	4 (44)		33 (3-NA)		NA (15-NA)	
Not Q61	8 (5)	5 (62)		17 (4-NA)		22 (8-NA)	
Unspecified	12 (7)	2(17)		5 (2–20)		33 (3-NA)	

(*) Exact Fisher Test; (**) Log Rank test, (IQR): Inter Quartile Range.

**Table 6 cancers-13-00475-t006:** Univariate and multivariate analysis of disease control rate including complete response, partial response and stable disease lasting more than 6 months.

Parameter	Univariate			Multivariate		
	OR	95% CI	*p* Value	OR	95% CI	*p* Value
Sex (M versus F)	1.09	(0.69–1.71)	0.72	-	-	-
Age (+1)	1.00	(0.99–1.02)	0.81	-	-	-
NRAS status (mut versus wt)	1.03	(0.66–1.60)	0.90	1.95	(1.07–3.54)	0.03
Site of primary melanoma			<0.001			0.05
Head & neck versus unknown	1.11	(0.45–2.71)		1.43	(0.42–4.85)	
Trunk versus unknown	0.53	(0.26–1.10)		0.50	(0.20–1.25)	
Upper limbs versus unknown	1.10	(0.42–2.88)		0.90	(0.28–2.92)	
Lower limbs versus unknown	0.31	(0.14–0.67)		0.37	(0.14–0.98)	
Other versus unknown	2.97	(0.33–26.4)		3.18	(0.29–35.2)	
AntiPD-1 versus ipilimumab	4.07	(2.39–6.95)	<0.001	5.81	(2.78–12.1)	<0.001
N. metastatic sites (<3 versus ≥3)	1.96	(1.25–3.07)	<0.01	2.64	(1.43–4.88)	<0.01
WBC (+1000)	0.95	(0.90–1.01)	0.10	-	-	-
Lymphocytes (+1000)	0.91	(0.77–1.08)	0.30	0.78	(0.55–1.11)	0.16
N/L ratio (+1)	0.88	(0.77–1.00)	0.05	0.91	(0.79–1.04)	0.17
Platelet (+100)	0.66	(0.50–0.87)	<0.01	0.67	(0.47–0.96)	0.03
LDH (<ULM versus >ULM)	1.73	(1.04–2.86)	0.03	-	-	-
ECOG PS (2–3 versus 0–1)	0.40	(0.24–0.67)	<0.01	0.40	(0.20–0.82)	0.01

**Table 7 cancers-13-00475-t007:** Univariate and Multivariable analysis of progression-free survival.

Parameter	Univariate			Multivariable		
	HR	95% CI	*p* Value	HR	95% CI	*p* Value
Sex (M versus F)	0.97	(0.74–1.27)	0.81	1.39	(0.99–1.96)	0.05
Age (+1)	1.01	(0.99–1.02)	0.34	-	-	-
NRAS status (mut versus wt)	0.92	(0.70–1.20)	0.53	0.73	(0.53–1.00)	0.05
Site of primary melanoma			0.01			0.03
Head & neck versus unknown	0.63	(0.37–1.06)		0.53	(0.27–1.04)	
Trunk versus unknown	1.02	(0.68–1.54)		1.12	(0.70–1.81)	
Upper limbs versus unknown	0.57	(0.32–1.03)		0.58	(0.30–1.14)	
Lower limbs versus unknown	1.30	(0.84–2.00)		1.16	(0.68–1.97)	
Other versus unknown	0.73	(0.28–1.88)		0.50	(0.17–1.51)	
AntiPD-1 versus ipilimumab	0.46	(0.34–0.61)	<0.0001	0.41	(0.29–0.60)	<0.0001
N. metastatic sites (<3 versus ≥3)	0.63	(0.49–0.83)	<0.01	0.60	(0.44–0.84)	<0.01
WBC (+1000)	1.01	(1.00–1.03)	0.15	-	-	-
Lymphocytes (+1000)	1.03	(0.92–1.16)	0.61	1.10	-	-
N/L ratio (+1)	1.04	(1.00–1.09)	0.06	1.05	-	-
Platelet (+100)	1.14	(0.96–1.35)	0.14	1.21	(1.01–1.44)	0.04
LDH (<ULM versus >ULM)	0.65	(0.48–0.89)	<0.01	0.78	(0.56–1.09)	0.15
ECOG PS (2–3 versus 0–1)	2.20	(1.65–2.93)	<0.0001	2.02	(1.42–2.86)	<0.0001

**Table 8 cancers-13-00475-t008:** Univariate and multivariate analysis of overall survival.

Parameter	Univariate			Multivariate		
	HR	95% CI	*p* Value	HR	95% CI	*p* Value
Sex (M versus F)	1.01	(0.75–1.37)	0.94	1.48	(1.00–2.18)	0.05
Age (+1)	1.00	(0.99–1.02)	0.64	-	-	-
NRAS status (mut versus wt)	0.82	(0.61–1.11)	0.20	0.72	(0.51–1.03)	0.07
Site of primary melanoma			0.03			0.01
Head & neck versus unknown	0.77	(0.43–1.40)		0.79	(0.34–1.82)	
Trunk versus unknown	1.11	(0.69–1.80)		1.62	(0.90–2.92)	
Upper limbs versus unknown	0.66	(0.34–1.28)		0.79	(0.34–1.82)	
Lower limbs versus unknown	1.45	(0.89–2.37)		2.11	(1.13–3.93)	
Other versus unknown	0.43	(0.10–1.82)		0.67	(0.15–3.04)	
AntiPD-1 versus ipilimumab	0.82	(0.59–1.13)	0.23	-	-	-
N. metastatic sites (<3 versus ≥3)	0.65	(0.48–0.87)	<.01	-	-	-
WBC (+1000)	1.01	(1.00–1.04)	0.03	1.02	(1.00–1.04)	0.04
Lymphocytes (+1000)	1.00	(0.89–1.11)	0.97	-	-	-
N/L ratio (+1)	1.06	(1.02–1.11)	<0.01	1.05	(0.99–1.11)	0.09
Platelet (+100)	1.26	(1.05–1.51)	0.01	1.31	(1.07–1.59)	<0.01
LDH_ (<ULM versus >ULM)	0.50	(0.36–0.69)	<0.001	0.57	(0.39–0.82)	<0.01
ECOG PS (2–3 versus 0–1)	1.77	(1.27–2.45)	<0.001	1.52	(1.02–2.25)	0.04

## Data Availability

Data available on request due to restrictions eg privacy or ethical The data presented in this study are available on request from the corresponding author. The data are not publicly available due to our institutional policy.

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
