# Peer review of "No Impact of NRAS Mutation on Features of Primary and Metastatic Melanoma or on Outcomes of Checkpoint Inhibitor Immunotherapy: An Italian Melanoma Intergroup (IMI) Study"

_cancers, 2021, doi:10.3390/cancers13030475_

Round 1
Reviewer 1 Report
The authors have gathered comprehensive retrospective data on patients with metastatic melanoma and analyzed the effect of first-line CII on patients with a NRAS mutation compared to patients with a wild-type NRAS status. Well done. They found that the cohorts in general were balanced despite minor differences. They did not find any difference in ORR, DCR, PFS or OS between the 2 groups and therefore conclude that their data do not support increased aggressiveness or higher responsiveness to CII for patients with NRAS mutated metastatic melanoma. The subject is relevant and would be interesting for the reader. It is important to publish studies with real-life populations and negative results.
The manuscript contains numerous data and analyses. It would be beneficial to reduce the amount of data and with less repetition of results as it is also presented in the tables. A reduction in test could be to exclude the data on the melanoma onset as the primary focus is on advanced disease and just mention that DFS is similar in the 2 cohorts.
The last line in the introduction is misleading. Is there a typing error?
In general, the method section needs a major revision. How are the patients selected? Consecutively? Time period? How are the NRAS analysis performed? From the primary tumour or metastasis? Are patients with a follow-up less than 3 months or patients without a response evaluation excluded if they die before 3 months? If so it will bias the data. Could they be included in the survival analyses? The PFS is very high for real-world patients which leads to the assumption that this is a highly selected population with a favourable risk profile (low disease stage and good PS). Have patients with the worst prognosis been excluded? A flowchart would be appreciated. What is the follow-up time?
For the result section
Is lower limb localization more frequent in mut/wt (41 pt/25%) compared to wt/wt (37/22%) as mentioned in the text?
There is something wrong with the numbers in site of disease in Table 3. Describe why it doesn´t equal 100% (probably due to numerous locations but should be described). The numbers/% for liver doesn´t make sense.
In table 5 the OS is less than PFS for "Not Q61". Can you explain?
In the text you interpret a modest benefit in terms of ORR for all patients ( 42% vs 37%, p=0.38). I would interpret it as no difference.
I find it concerning that NRAS is non-significant in the univariate analysis and still entered into the multivariate analyses were it becomes significant. You mention that there is interaction but between which variables? You should control for interaction between these variables in the multivariate analysis. What happened to the other non-significant variables from the univariate analyses? Where they also entered in the multivariate analyses but not significant in the stepwise selection? If so how did you define the parameters that you put into the uni/multivariate analyses? If unselectively, how come you didn´t analyze performance status as it is one of the strongest parameters and you have it? I could only find the referred article 14 in chinese.
You conclude that your data do not support higher responsiveness for CII in NRAS mutated MM as you find no difference in DCR, ORR, PFS or OS but your applied multivariate analyses describes NRAS mutation as an independent factor significantly associated with better DCR and PFS and borderline significant for OS irrespectively of other risc factors. I do not agree with your strict conclusion. I don´t find that you data gives a clear understanding of NRAS aggressiveness/responsiveness as there is no difference in effect parameters but the multivariate analyses indicate that NRAS influence DCR, PFS and OS (borderline) but I find the results from the multivariate analyses uncertain. The data put another aspect on NRAS but not clearly. There are still to many uncertainties with retrospective data which is still highly needed. Bias should be described more clearly.
Reviewer 2 Report
It was considered that NRAS-mutant melanoma is more aggressive and more responsive to checkpoint inhibitor immunotherapy when compared to NRAS wildtype. The authors retrospectively recruited 331 metastatic melanoma patients treated with checkpoint inhibitor immunotherapy as first line: 162 NRAS-mutant/BRAF wild-type and 169 wt/wt. No substantial differences were observed among the two cohorts regarding the melanoma onset and disease-free interval. Also, overall response to checkpoint inhibitor immunotherapy, progression-free survival and overall survival were similar in the two groups. Therefore, their data do not support increased aggressiveness and higher responsiveness to checkpoint inhibitor immunotherapy of NRAS-mutant melanoma. This is an excellent study with all appropriate controls with great clinical importance in the field of melanoma.
Reviewer 3 Report
The authors presented a retropective, multicentric analysis evaluting a timely and interesting topic on response to ICC in pts with NRAS mutated melanoma.
The number of patients included is elevated and allows for some general conclusions.
The manuscript could benefit from being reviewed by an English native speaker.
Some more detailed comments can be found next.
Simple summary and conclusions: “We believe our paper definitively cleared up these controversial issues”- please remove the sentence as this statement cannot be done based in one retrospective review albeit multicentric.
Abstract: Please define the abbreviations when used the first time: “Regarding the outcomes to CII, no significant differences were reported in ORR, DCR, PFS or OS” “lower N/L ratio”
“Therefore, the standard treatment of NRAS-mutant patients is currently the same as for BRAF wild-type melanoma; anti-PD-1 immunotherapy is the first-line therapy (…)” - Consider changing to “Therefore, the standard treatment of NRAS-mutant patients is currently the same as for BRAF wild-type melanoma; anti-PD-1 based immunotherapy is the first-line therapy (…)”
Table 1 – please define symbol after therapy second line “yes” 56.
Page 10: regarding the study from Jonson and colleagues – “Of note, no survival data were reported in this study.” The authors want to refer to OS? As PFS data was actually presented. Consider correction for accuracy.
Page 11: Regarding the anti-PD-1 group, we reported a PFS longer than in large controlled studies which is of approximately 5-7 months [4,21]. This longer PFS could be explained by the better prognosis of our patient population (LDH approximately 30% vs. over 35% and M1c stage of approximately 45% vs. approximately 60%) – the authors meant Normal LDH?
Page 12: However, some weaknesses remain and should be highlighted, including the retrospective nature of the study. – please elaborate more on the shortcomings of the study.
A significant part of the discussion is taken with a description of the results again. Please consider reviewing and shortening mention to results again, when not applicable and where no new information is given besides the one already in. the tables and the results section.
Reviewer 4 Report
This manuscript explains a current topic therefore of interest to the readers. This is a clear, concise and well written manuscript. The subject matter is relevant and complete. The methods are generally appropriate. Overall, the results are clear and supported by careful statistical analysis that confirm high scientific soundness. The authors make a systematic contribution to the research literature.
